# Adaptation of the Brain to Hyponatremia and Its Clinical Implications

**DOI:** 10.3390/jcm12051714

**Published:** 2023-02-21

**Authors:** Fabrice Gankam Kengne

**Affiliations:** Nephrology Division, EpiCURA Hospital, Rue Maria Thomée 1, 7800 Ath, Belgium; gankam@gmail.com

**Keywords:** hyponatremia, osmolarity, osmotic demyelination, brain, astrocytes

## Abstract

Hyponatremia is the most common electrolyte disorder, occurring in up to 25% of hospitalized patients. Hypo-osmotic hyponatremia when severe and left untreated invariably results in cell swelling, which can lead to fatal consequences, especially in the central nervous system. The brain is particularly vulnerable to the consequences of decreased extracellular osmolarity; because of being encased in the rigid skull, it cannot withstand persistent swelling. Moreover, serum sodium is the major determinant of extracellular ionic balance, which in turn governs crucial brain functions such as the excitability of neurons. For these reasons, the human brain has developed specific ways to adapt to hyponatremia and prevent brain edema. On the other hand, it is well known that rapid correction of chronic and severe hyponatremia can lead to brain demyelination, a condition known as osmotic demyelination syndrome. In this paper, we will discuss the mechanisms of brain adaptation to acute and chronic hyponatremia and the neurological symptoms of these conditions as well as the pathophysiology and prevention of osmotic demyelination syndrome.

## 1. Introduction

In cell physiology, a semipermeable membrane describes a membrane separating two compartments that is permeable only to certain substances and non-permeable to others.

Osmosis describes a phenomenon in which when two compartments are filled with a solvent and are separated by a semipermeable membrane, the movement of the solvent across the membrane is driven by the concentration of the solute (osmotic concentration or osmolarity) in the respective compartment. The solvent will move across the membrane from the compartment with the lowest osmolarity to the compartment with the highest osmolarity. The driving force that makes this movement possible is called the osmotic pressure, which depends mainly on the concentration of solutes across the membrane. The compartment with the highest osmolarity will have a higher osmotic pressure, while the compartment with the lowest osmolarity will have a lower osmotic pressure. Water will then move from the compartment with the lower osmotic pressure to the compartment with the higher osmotic pressure until the osmotic pressure in both compartments is the same [1].

Another important phenomenon that underlies movements across a semipermeable membrane is the Gibbs–Donnan effect. It describes the movement of charged particles across the semipermeable membrane in order to maintain electroneutrality across the membrane or if not to reach an equilibrium called Gibbs–Donnan equilibrium at which no movement of charged particles is seen.

The cell membrane, which separates the inside of the cell (intracellular compartment) from the outside of the cell (extracellular compartment), is made of a phospholipid bilayer, which from a physiological standpoint serves as a semipermeable membrane. This means that the cell membrane is only selectively permeable to some solutes. For review, see [2,3].

Both the intracellular and extracellular compartments are filled with solutes of different natures, many of them being charged particles such as ions and proteins. Movement of solutes across the cell membrane can occur through various mechanisms, including passive diffusion, active transport, and passive transport. Movement of water across the cell membrane occurs mainly through water channels called aquaporins [4].

The intracellular compartment is selectively enriched with large anionic molecules (mostly proteins). The cell membrane is impermeable to these molecules, and therefore other small diffusible charged molecules can move in order to achieve the Gibbs–Donnan equilibrium. This process will invariably modify the osmotic pressure across the membrane and drive water movement.

Movement of water across the cell will affect the global water content of the cell. In extreme cases, when there is too much water inside the cell, it will burst, and conversely, when there is too little water inside the cell, it will shrink. Early application of these principles was provided by the work of Hewson on hemolysis where, when red blood cells were submitted to a hypo-osmotic solution, they exploded (a phenomenon called hemolysis), and likewise they shrunk when immersed in a hyperosmotic solution [5].

Serum sodium concentration is the main determinant of extracellular osmolarity. Serum sodium concentration is determined in turn by using the Edelman equation, which is the sum of total exchangeable body sodium and potassium over the total body water volume [6].

Hyponatremia is a very common disorder that has diverse etiology. It can be classified depending on the etiology based on Edelman’s equation. The plasma concentration of sodium will be decreased if (i) the total body water is increased (overload hyponatremia), (ii) the sum of the exchangeable sodium or potassium is decreased (depletion hyponatremia), or (iii) there is a decrease in the sum of exchangeable total serum sodium and potassium along with a proportionally smaller decrease in the total body water.

Hyponatremia can also be classified based on its duration, with a drop in serum sodium for less than 48 h being considered acute whereas after 48 h it is chronic.

Finally, depending on the severity of hyponatremia (i.e., the magnitude of serum sodium drop) hyponatremia can be classified as mild, moderate, or severe. It is worth mentioning though that this classification can be confusing because there is no uniform agreement on the normal serum sodium value. When graphing the mortality against the serum sodium value in an ambulatory cohort, we have previously shown that the serum sodium level at which the minimal mortality is seen is around 137 mEq/L [7]. Another study in hospitalized patients found that the serum sodium value associated with the lowest mortality was around 140 mEq/L [8]. In contrast, many laboratories still report a “normal” serum sodium range at 134 mEq/L. Thus, the cutoff used to define mild, moderate, and severe hyponatremia can vary, and this will invariably affect the incidence of these disorders.

Regardless of the cutoff used to define hyponatremia, its incidence significantly varies depending on the studied population, with the highest incidence invariably seen in hospitalized patients and the lowest incidence in the community.

A large study reported that general prevalence of hyponatremia defined as a serum sodium level below 135 mEq/L can reach up to 28% in hospitalized patients, while severe hyponatremia defined as a serum sodium lower than 116 mEq/L occurred in 0.49%, 0.17%, and 0.03% of hospitalized, ambulatory, and in community patients, respectively [9].

Another study only in hospitalized patients found that a serum sodium of less than 120 mEq/L was found in 0.2% of the studied population [10].

## 2. Brain Adaptation to Acute Hyponatremia

It is expected that a drop in the serum sodium concentration will result in a proportional drop in the extracellular osmolarity. In that case, water will flow from the extracellular compartment to the inside of the cell. This will cause dilution of the intracellular compartment. In extreme cases as discussed before, cellular edema and cell rupture may ensue.

The adult brain is encased in a rigid structure that limits its expansion. If the brain volume increases too much, herniation of the brain will occur through defined anatomical areas.

Brain cells have developed significant mechanisms to prevent cell volume increase during hyponatremia. These mechanisms are collectively called mechanisms of regulatory volume decrease as they counter-regulate the increase in cell volume imposed by hypo-osmolarity [11,12].

These mechanisms occur in a time-defined fashion [13], the first being the release of intracellular potassium and chloride, which will decrease intracellular ionic content and induce an osmotic shift of the water from the inside of the cell to the outside, in order to prevent further cell volume increase. This is seen within minutes of hypo-osmotic cell swelling and is maximal after 3 h [14,15,16].

Another mechanism that occurs after the first one involves extrusion of non-ionic osmotically active substances from the inside of the cell. They are mostly organic osmolytes of diverse natures, including betaine, taurine, and myo-inositol as the most important [17,18,19]. This phenomenon might be mediated by active transport involving an increase in the activity and/or an increase in the number of transporters [20,21].

Experimental studies as well as clinical observation have shown that despite these adaptive mechanisms, hyponatremia might still induce symptoms attributed to cell swelling [22]. This suggests that the buffering capacity of the brain cell is unfortunately not perfect. In other terms, there is a threshold at which the regulatory volume decrease mechanisms are insufficient to prevent cell swelling. This depends on both the amplitude and the acuteness of hyponatremia as the mechanisms of regulatory volume decrease are sequential. Indeed, in acute and profound hyponatremia the content of intracellular ions (K, Na, and Cl) will be insufficient to counterbalance the water influx inside the cell, especially if hyponatremia developed at a rate that prevented the proper installation of organic osmolyte extrusion [22,23,24,25]. Likewise, the buffering capacity of organic osmolytes is also limited, and if the cause of hyponatremia persists after maximal extrusion of organic osmolytes, cell swelling can still occur as evidenced by symptoms seen in acute on chronic hyponatremia [23].

## 3. Symptoms of Acute Hyponatremia

Acute hyponatremia can induce a myriad of neurological symptoms depending on the severity [22,25,26,27]. Previous studies have suggested that the symptoms could range from mild cognitive impairment to seizures and even brain herniation and death (see Table 1).

It is however very difficult to put a threshold on what is considered acute hyponatremia in terms of time of installation. Most experts agree that acute hyponatremia develops within 48 h. This practical definition was adopted because it is considered that the secondary mechanisms of RVD (organic osmolyte extrusion) are fully in place after 48 h. Still, in a clinical setting when facing a patient in the emergency department or in a ward with hyponatremia, in many instances serum sodium value in the preceding hours might be lacking. For these reasons, it is our opinion that symptomatic hyponatremia should be considered acute or acute on chronic until proven otherwise.

## 4. Treatment of Acute Hyponatremia

The main goal of treatment for acute hyponatremia is to prevent further cell swelling, which will cause brain herniation. To achieve this goal, extracellular (plasma) osmolarity should be increased until the symptoms stop. Several algorithms have been proposed to achieve such a goal. Based on neurosurgery studies, it has been determined that in most cases, an increase in serum sodium of 5–6 mEq/L will be enough to decrease brain volume to prevent brain herniation [28,29]. Such an increase can be obtained through serial boluses of hypertonic saline, and some authors have suggested that enteral urea could be useful in decreasing intracranial pressure in hyponatremic brain injury patients [30].

In any case, serum sodium should be increased enough to stop the symptoms.

Table 2 and Figure 1 represent a proposed treatment approach for severe hyponatremia. Figure 1 is an algorithm adapted from a consensus among European experts for treatment of acute symptomatic hyponatremia [31].

## 5. Brain Adaptation to Chronic Hyponatremia

When hyponatremia develops over days and at a slow pace, brain cells have enough time to put in motion more sustained mechanisms of adaptation. It has been shown that brain water content in chronically hyponatremic animals is roughly the same as in normonatremic animals [24,32,33]. Considering that the maximal outflow of ions in the brain is achieved within a few hours of hyponatremia installation, this evidence suggests that other mechanisms must have taken place to maintain normal water balance in the brain. As discussed above, these mechanisms involve the extrusion of several organic osmolytes from the intracellular compartment. The most studied brain organic osmolytes are betaine, taurine, glutamate, and myo-inositol. Several studies have confirmed that the brain content of these substances in hyponatremic animals and humans is strikingly decreased [17,18,34]. It is still unclear whether or not the kinetics of brain depletion of organic osmolytes is uniform in terms of anatomical distribution, and some studies have shown that there are some areas of the brain that are more prone to organic osmolyte loss [35,36]. Likewise, it is unknown whether or not these organic osmolytes are interchangeable in their osmotic brain buffering capacity.

One might think that since the brain volume in chronic hyponatremia remains stable, chronic hyponatremia should not induce any significant clinical consequences. That is not true as several experimental and clinical studies have now showed that chronic hyponatremia is associated with significant neurological impairment, albeit less striking and dramatic than in acute hyponatremia.

## 6. Symptoms and Treatment of Chronic Hyponatremia

A particularly well conducted animal study showed that chronic hyponatremia in rats induces significant gait disturbance and memory impairment, and mechanistically it was hypothesized that chronic hyponatremia induces significant changes in synaptic plasticity, partly because of depletion of brain glutamate content [37]. Prior to this experimental observation, it was already well established that chronic hyponatremia in humans could induce gait disturbances and significant attention deficit, which were corrected upon normalization of serum sodium [38].

Other well described neurological manifestations of chronic hyponatremic encephalopathy are less dramatic than in acute hyponatremic encephalopathy, and these include malaise, nausea, gait deficit, attention disturbances, and mild confusion [18,38,39].

Falls and bone fractures are also more frequently seen in patients with chronic hyponatremic encephalopathy [38,40]. Brain imaging is usually normal with no signs of brain edema.

Table 1 compares and contrasts neurological manifestations of acute and chronic hyponatremia.

The treatment of chronic hyponatremia usually involves treating the cause. In the absence of acute neurological manifestations, specifically seizures or severely altered mental state, treatment of chronic hyponatremia should be undertaken with much caution and over several days as rapid correction of chronic hyponatremia could lead to osmotic demyelination syndrome (ODS).

An important parameter to consider when treating chronic hyponatremia is the duration of hyponatremia. Usually, patients with fully adapted chronic hyponatremia will have no symptoms of brain edema, and those patients should be corrected very slowly in the appropriate setting with readily available serial serum sodium measurements when the chronic hyponatremia is severe.

Although there is abundant evidence that associates poor clinical outcomes with even mild clinical hyponatremia, it remains unclear whether meaningful clinical benefit can be derived from the treatment of mild chronic hyponatremia (serum Na > 132 mEq/L). From a neurological standpoint, patients with a serum Na of >128–130 mEq/L are at a lower risk of brain complications during correction of hyponatremia. Figure 1 provides a treatment scheme for chronic hyponatremia.

Several agents have been shown to be effective in the treatment of chronic hyponatremia depending on the cause. These include urea (for SIADH), vaptans, fluid restriction, and salt tablets.

## 7. Osmotic Demyelination Syndrome (ODS)

When chronic and severe hyponatremia is corrected rapidly in humans, it has been shown that in some cases brain damage can ensue [41,42,43]. Experimental studies have confirmed that chronicity of hyponatremia as well as the magnitude and speed of correction are the most important risk factors for development of myelinolysis during correction of serum sodium [44,45,46,47,48]. Brain imaging, when obtained after 2–3 days following correction of the hyponatremia, will usually show radiological signs of myelin loss in various areas of the brain, including but not limited to the pons [49,50,51]. In fact, extrapontine lesions are at least as common as pontine lesions, which are rarely isolated [52]. This syndrome is called osmotic demyelination syndrome, and sometimes the term central pontine myelinolysis is used as historically it was believed that only the central pontine region was affected.

ODS is a very rare phenomenon and can also occur in instances other than rapid correction of hyponatremia, such as severe hypernatremia. The term central pontine myelinolysis is often used to described demyelination not related to osmotic disturbances. 

The incidence of central pontine myelinolysis has been reported to be around 0.61 per million, with as much as 86% being associated with correction of hyponatremia [53].

## 8. Physiopathology of ODS

The study of an animal (rat) model of osmotic demyelination syndrome has allowed us to gain much insight into the mechanisms leading to demyelination after rapid correction of chronic hyponatremia.

From the most recent studies, it is now clear that demyelination is only the final part of a complex pathological process.

Several factors play a role in the pathophysiology of the disease.

Historically, the rupture of the blood–brain barrier (BBB) was believed to be the main culprit allowing myelinotoxic substances such as complement factors to exert deleterious effects on the brain [54,55]. Subsequent studies have shown that despite a nearly intact BBB permeability, demyelination lesions can still ensue. This has clarified that the opening of the BBB is probably a two-step phenomenon with one brief osmotic opening and a second more lasting inflammatory disruption, which occurs as a consequence of the demyelination itself [56,57].

Likewise, microglial activation was previously thought to be cardinal in the mechanisms leading to demyelination, but now it is accepted that earlier and more significant events take place before microglial activation [58].

Astrocytes are known to play a significant part in the pathophysiological process with both astrocyte hypertrophy and gliosis as well as astrocyte death [56,59,60]. In fact, astrocyte death was shown to occur very early in regions that will later be demyelinated, whereas astrogliosis actually occurs later around the demyelinating regions [58,61]. A later study suggested that rapid correction of chronic hyponatremia induces an increase in the amount of misfolded and insoluble proteins, which overload the processing capacity of the astrocyte, leading to cell death [59]. It is believed that organic osmolytes that are depleted during chronic hyponatremia and not fully replenished during its correction could help to prevent osmotic and ionic buffering abilities of the cells and decrease the unfolded protein response and endoplasmic reticulum stress in astrocytes [59] (see Figure 2).

Rapid correction of hyponatremia with urea or in uremic animals rarely results in osmotic demyelination [62,63], and it has been hypothesized that urea can act as an organic osmolyte. Likewise, administration of exogenous myo-inositol was able to prevent osmotic demyelination after rapid correction of chronic hyponatremia.

## 9. Clinical Course and Prognosis of Osmotic Demyelination

The clinical course of ODS is variable. Some cases of osmotic demyelination follow a biphasic course where there is an initial improvement of the clinical status of the patients as chronic severe hyponatremia is corrected, followed by a rapid deterioration when demyelination actually sets in, which is usually 2–3 days after the initial correction. Other cases evolve directly into severe neurological impairment without the initial improvements.

Once believed to be invariably fatal, recent reports have shown that the prognosis is actually very variable with as many as half of the patients experiencing significant recovery [64,65,66].

## 10. Treatment of Osmotic Demyelination

Several experimental approaches have been suggested for the treatment of osmotic demyelination including administration of steroids, plasmapheresis, and minocycline. However, to date the best approach relies on prevention of too rapid correction of chronic hyponatremia, and if inevitable, relowering of serum sodium has been shown to be helpful both experimentally and clinically when undertaken in the 24 to 48 h following the initial overcorrection [67,68,69]. Such an approach involves administration of desmopressin along with either saline or dextrose infusion and should be undertaken with caution under expert guidance.

## 11. Conclusions and Perspectives

The brain as an organ has developed extraordinary adaptation mechanisms to counterbalance the effects of hyponatremia and hypo-osmolarity. However, these mechanisms are clinically challenged in daily clinical practice. Indeed, the neurological consequences of hyponatremia and its correction still carry a significant toll in terms of mortality and morbidity.

A tremendous amount of work has been accomplished in terms of improving our understanding of the physiopathology of hyponatremic encephalopathy, either acute or chronic, and the mechanisms of osmotic demyelination syndrome.

There are some simple rules and protocols that can be followed to minimize the deleterious consequences of hyponatremia or its correction on the brain, and every clinician should be familiar with them. Although there are still many research questions that remain unsolved regarding the neurological complications of hyponatremia and its treatment, careful understanding of the physiological basis of brain response to hypo-osmolarity can certainly lead to a better prevention of these deleterious neurological consequences.

## Figures and Tables

**Figure 1 jcm-12-01714-f001:**
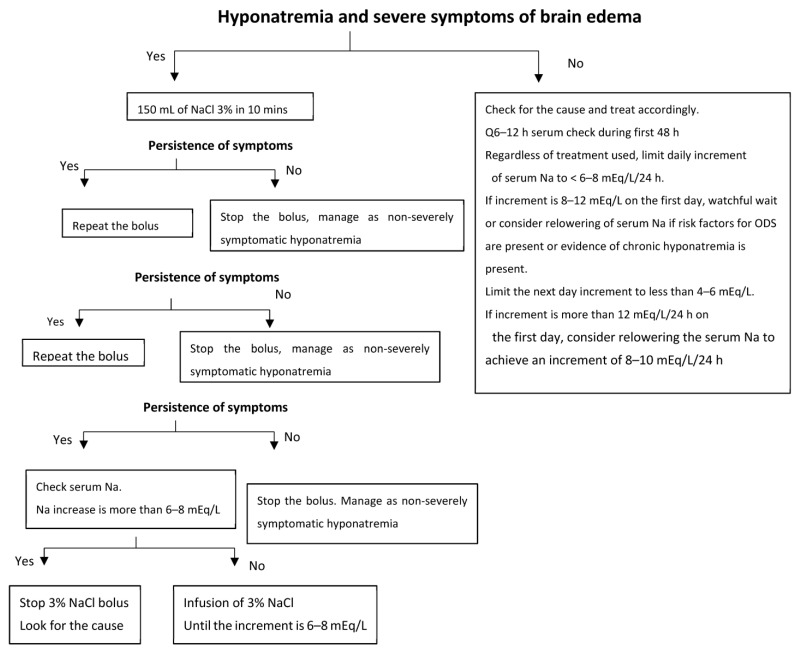
Proposed algorithm for management of hyponatremia with regards to brain complications. The most important parameter in determining the need for urgent treatment should be the presence of neurological symptoms attributable to hyponatremia and not the chronicity of hyponatremia or the magnitude of hyponatremia. Chronic hyponatremia with no neurological symptoms is a risk factor for ODS and dictates very slow correction of serum sodium, regardless of the correction method selected. Acute hyponatremia with no neurological symptoms should not be corrected rapidly as the assessment of the acuteness of hyponatremia can be biased, which may pose a risk for ODS. The limits of serum sodium increments are based on the current state of the literature. Severe hyponatremia with neurological symptoms should be approached regardless of its cause. Reproduced from [9]: Kidney International Reports, 2017 with permission.

**Figure 2 jcm-12-01714-f002:**
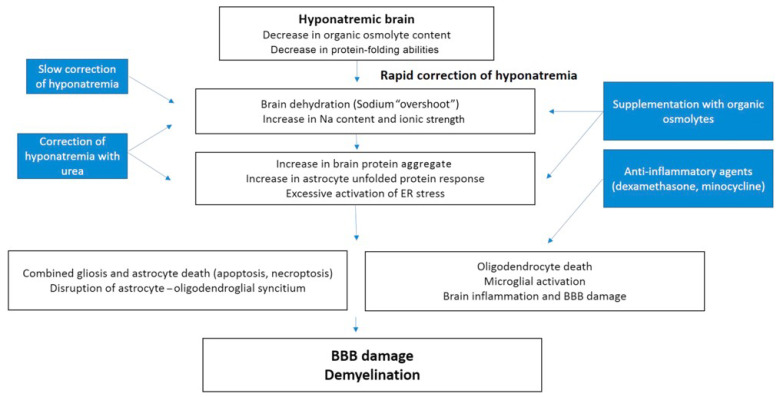
Proposed model for the physiopathology of osmotic demyelination syndrome. Reproduced from [9]: Gankam et al.: Kidney International Reports, 2017 with permission. BBB: blood–brain barrier, ER: endoplasmic reticulum.

**Table 1 jcm-12-01714-t001:** Manifestations of hyponatremic encephalopathy.

Acute Hyponatremia	Chronic Hyponatremia
Nausea and vomiting	Nausea
Headaches	Fatigue
Seizures	Gait and attention deficit
Coma and death	Falls and bone fractures
Respiratory arrest Non-cardiogenic pulmonary edema	

Reproduced from [9]: Gankam et al.: Kidney International Reports, 2017 with permission.

**Table 2 jcm-12-01714-t002:** Proposed approach for the treatment of severe hyponatremia.

Treatment of Severe Hyponatremia	
**Signs or symptoms of Brain Edema**	**No signs or symptoms of brain edema**
ICU admission Hypertonic saline boluses (100 mL NaCl 3%) Repeat if symptoms persist Measure serum Na q1–2 h Stop when symptoms abate Stop when sodium increment is 6 mEq/L Keep checking serum Na q3–4 h	Etiological evaluation Determine chronicity (>48 h) Assess for risk factors for ODS (chronic hyponatremia, hypokalemia, liver disease, alcoholism) Treatment of the cause and avoid hypertonic saline If risk factors for ODS are present q6–12 h serum Na check Limit increment of Na to < 6 mEq/L/24 h * If no risk factors for ODS Limit increment of Na to < 8 mEq/L/24 h * Additional measures: If SIADH, use of urea is better Relower the serum Na if increment of > 12 mEq/L/day Desmopressin + D5W

ODS: osmotic demyelination syndrome, SIADH: syndrome of inappropriate antidiuretic secretion, D5W: dextrose 5% in water, Na: sodium. * These limits are based on the current state of the literature.

## Data Availability

Not applicable.

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
