# Peer review of "Adaptation of the Brain to Hyponatremia and Its Clinical Implications"

_jcm, 2023, doi:10.3390/jcm12051714_

Round 1

Reviewer 1 Report

Valuable topic. English editing required. Additional paragraph(s) are needed to better define categories of hyponatremia (related to the volume status of the patient) that best benefit from the administration of hypertonic saline solutions.

Author Response

We will like to thanks the reviewers for theirs insightfulm comments and their time.

We have performed the language editing.

Regarding the additional paragraph, we have included a paragraph that explains the classification of hyponatremia.

as for the type of hyponatremia that benefits from administration of hypertonic saline, in the acute setting, as long as the hyponatremia and symptomatic, hypertonic saline will work. we have adapted the text to reflect that.

Reviewer 2 Report

REPORT FROM THE REVIEWER

TITLE: Adaptation of the brain to hyponatremia and its clinical implications.

First, I would like to give thanks to the Editorial team of JCM for giving to me the opportunity to review the draft.

I want also to congratulate to the authors for performing a so much relevant review regarding this topic. All suggestions below should be considered as major points for pondering.

General considerations

- If the authors start giving an incidence of hospital hyponatremia nearly to 25%, obviously they are talking about all hyponatremias: mild, moderate and severe. Nevertheless when the whole manuscript is read, one can release that it is about only severe cases of hospital hyponatremia. For this reviewer the proper way to proceed is also giving the incidence of severe cases of hyponatremia (Authors are suggested to read an article recently published precisely in this same journal, JCM, it is: (Rius-Peris JM, Tambe P, Chilet Sáez M, Requena M, Prada E, Mateo J.  Incidence and Severity of Community- and Hospital-Acquired Hyponatremia in Pediatrics. J Clin Med. 2022 Dec 19;11(24):7522. doi: 10.3390/jcm11247522. PMID: 36556138). Certainly, this original research is centered only in pediatric patients, but it is a good and recent report regarding the true incidence of the phenomenon of hospital hyponatremia. This, as a severe form, can be a very dramatic clinical feature from the point of view of its consequences, but extremely rare. Then, the authors can not give so much relevance to the phenomenon under review (incidence of 30%) because they are reviewing a topic that means probably only less than 1% of cases of hyponatremia. I suggest a modification in the text (abstract) and add this argue in the introduction section  and also add this cite in bibliography section.

A general suggestion or advice for the authors: one phenomenon under study or under review must be always qualitative and quantitative considered in the introduction of any original or review article. That is very important for the potential readers.

Specific considerations

Line 29. It is written: “… from the compartment with the osmolarity to the compartment with the highest osmolarity.…” but must be written: “… from the compartment with the lowest osmolarity to the compartment with the highest osmolarity.…”

Line 33-35. What does the authors mean with?  “….Water will then move from the compartment with the highest osmotic pressure to the compartment with the lowest osmotic pressure until the osmotic pressure in both compartments is the  same…” Does not highest osmotic pressure mean compartment with highest osmolarity? Then, there is a mistake because the solute or the water goes from the compartment of lowest osmolarity to compartment of highest osmolarity. Read this paragraph in the manuscript again and correct it if is necessary.

Line 60. It is written: “when the is too little water inside the cells…” must be written: “when there is too little water inside the cells…”

Line 131, “proposed” instead of “proposaed”.

Line 203. What is that of unpublished evidence? What does the author mean? I suggest to give a reference regarding that in bibliography section.

Line 287 “…hyponatremia encephalopathy either chronic or chronic and the mechanisms of…” Correct please, chronic is written twice.

I don’t know the meaning of: RBC (line 61); RVD first time in line 94; ER, I supposed that means Emergency Room, but must be indicated clearly by the authors.

No more modifications suggested on the main text.

Regarding the tables and figures

The tables and figures contrast with the rest of the text from an aesthetic point of view.

Letter size and style in table 1 and 2 is not in accordance with letter size of the text in the different paragraphs along the manuscript.

Authors report that table 1 is adapted from Gankam et al. but what is about table 2? Is all information proposed on table 2 as a result of the work of reviewing of the authors or it was taken from another researcher?

Figures 1 and 2 are not adapted from Gankam et al. These figures are just exactly copied from a report published for previous mentioned researcher and this is not correct or even if legal. THIS QUESTION MUST BE JUDGED FROM ETITORIAL TEAM.

No further suggestions in any other section of the draft

Round 2

Reviewer 2 Report

Thanks to the authors for their quick corrections and respond.

Now, the manuscript is fine.